# Identification and Validation of PTGS2 Gene as an Oxidative Stress-Related Biomarker for Arteriovenous Fistula Failure

**DOI:** 10.3390/antiox13010005

**Published:** 2023-12-19

**Authors:** Ke Hu, Yi Guo, Yuxuan Li, Shunchang Zhou, Chanjun Lu, Chuanqi Cai, Hongjun Yang, Yiqing Li, Weici Wang

**Affiliations:** 1Department of Vascular Surgery, Union Hospital, Tongji Medical College, Huazhong University of Science and Technology, Wuhan 430000, China; 2023xh5014@hust.edu.cn (K.H.); d202282268@hust.edu.cn (Y.G.); m202175907@hust.edu.cn (Y.L.); chanjun_lu@hust.edu.cn (C.L.); chuanqicai@hust.edu.cn (C.C.); 1992xh0544@hust.edu.cn (Y.L.); 2Center of Experimental Animals, Huazhong University of Science and Technology, Wuhan 430000, China; 1989020590@hust.edu.cn; 3Key Laboratory of Green Processing and Functional New Textile Materials of Ministry of Education, Wuhan Textile University, Wuhan 430200, China; hjyang@wtu.edu.cn

**Keywords:** arteriovenous fistula failure, oxidative stress, prostaglandin-endoperoxide synthase 2

## Abstract

(1) Background: Arteriovenous fistulas (AVFs) are the preferred site for hemodialysis. Unfortunately, approximately 60% of patients suffer from AVF failure within one year. Oxidative stress plays an important role in the occurrence and development of AVF. However, the underlying mechanisms remain unclear. Therefore, specific oxidative stress-related biomarkers are urgently needed for the diagnosis and treatment of AVF failure. (2) Methods: Bioinformatics analysis was carried out on dataset GSE119296 to screen for PTGS2 as a candidate gene related to oxidative stress and to verify the expression level and diagnostic efficacy of PTGS2 in clinical patients. The effects of NS398, a PTGS2 inhibitor, on hemodynamics, smooth muscle cell proliferation, migration, and oxidative stress were evaluated in a mouse AVF model. (3) Results: Based on 83 oxidative stress-related differentially expressed genes, we identified the important pathways related to oxidative stress. PTGS2 may have diagnostic and therapeutic efficacy for AVF failure. We further confirmed this finding using clinical specimens and validation datasets. The animal experiments illustrated that NS398 administration could reduce neointimal area (average decrease: 49%) and improve peak velocity (average increase: 53%). (4) Conclusions: Our study identified PTGS2 as an important oxidative stress-related biomarker for AVF failure. Targeting PTGS2 reduced oxidative stress and improved hemodynamics in an AVF mouse model.

## 1. Background

In 1966, Brescia et al. succeeded in inventing a mature arteriovenous fistula (AVF) by surgically connecting an artery to a vein, thereby providing end-stage renal disease (ESRD) patients with dialysis access capable of delivering optimal flow [1]. Compared with other vascular access methods, arteriovenous fistulas have better long-term patency rates, lower complication rates, and lower medical costs and are recommended as the first choice of vascular access for renal replacement therapy (RRT) [2,3]. However, the patency rate of AVF was 60% 1 year after surgery [4]. Intimal hyperplasia of the outflow tract severely restricts the clinical applications of AVF. The exact mechanism underlying AVF failure is not fully understood. Therefore, an improved understanding of the molecular mechanisms and pathophysiology of AVF failure will aid in the development of new diagnostic and therapeutic strategies and improve clinical outcomes.

Strong evidence suggests that oxidative stress plays a significant role at every stage of AVF maturation [5,6,7], including preoperative underlying mechanisms, intraoperative surgical damage, and postoperative hemodynamic alterations. Oxidative stress occurs when there is an imbalance between the generation of reactive oxygen species (ROS) and the antioxidant defense mechanisms of the body, referred to as the “redox state”. At relatively low levels, ROS aid in maintaining stability within the cellular environment by acting as secondary messengers. However, excess ROS impair this redox equilibrium, leading to damage to proteins and DNA, lipid peroxidation, irreversible cellular damage, and death [8].

In this study, we employed a variety of bioinformatics approaches to discover that prostaglandin endoperoxide synthase 2 (PTGS2) may play a key role in the occurrence and development of AVF failure and may be a potential oxidative stress-related biomarker for this condition. PTGS2 is the human gene encoding cyclooxygenase 2, which is a key enzyme mediating inflammation in vivo. It is induced by a variety of factors, such as cytokines, growth factors, and tumor-promoting factors, and plays an important role in inflammation and many cardiovascular diseases [9,10]. Several studies have demonstrated the protective effect of PTGS2 inhibition in vascular smooth muscle proliferative diseases, including vascular injury and pulmonary hypertension [11,12]. Several studies have explored the molecular mechanisms of PTGS2 that are involved in the proliferation, migration, and phenotypic transformation of smooth muscle cells [13,14,15]. These results suggest that PTGS2 plays a key role in intimal hyperplasia-induced AVF stenosis. The inhibition of PTGS2 may be an important intervention for the prevention and treatment of AVF failure.

This study identified the diagnostic genes for AVF failure using a bioinformatics approach incorporating oxidative stress and validated them using an additional external dataset. We subsequently found that PTGS2 expression was increased in failed AVF samples obtained from patients. In a mouse model of AVF, administration of the selective PTGS2 inhibitor NS398 reduced intimal hyperplasia and improved hemodynamics in the outflow tract vessels. PTGS2 inhibitors also affect the function and status of smooth muscle cells and fibroblasts and ameliorate oxidative stress (as shown in Appendix A for the specific study route).

## 2. Methods

### 2.1. Data Source

The GSE119296 dataset of AVF patients was extracted from the NCBI Gene Expression Omnibus (GEO) database (https://www.ncbi.nlm.nih.gov/geo/, accessed on 15 February 2023). Raw bioinformatic RNA-seq gene counts were obtained from GSE119296 using the GPL18573 platform (Illumina NextSeq 500, San Diego, CA, USA) [16]. Seven native veins and seven AVFs with failed maturation outcomes were collected from the venous segments of AVFs in hemodialysis patients. Read counts were also converted to fragments per kilobase of transcripts per million mapped reads (FPKMs) according to gene length. Additionally, using a relevance score greater than 5 as a screening criterion, we extracted 1760 genes associated with oxidative stress from the Genecard database (version: 5.17; https://www.genecards.org/; accessed on 10 February 2023).

### 2.2. Identification of Oxidative Stress-Related Differentially Expressed Genes (OSDEGs)

We screened differentially expressed genes (DEGs) from GSE119296 using the R language package “limma” (version 3.58.1; https://bioconductor.org/packages/release/bioc/html/limma.html; accessed on 15 January 2023), with |log2 fold change (FC)| > 0.5 and *p* < 0.05 as screening criteria [17]. Subsequently, the obtained DEGs were intersected with 1760 oxidative stress-related genes to obtain oxidative stress-related differentially expressed genes (OSDEGs).

### 2.3. Functional Enrichment Analysis of OSDEGs

Gene Ontology (GO) and Kyoto Encyclopedia of Genes and Genomes (KEGG) enrichment analyses were conducted to analyze the functional enrichment of OSDEGs associated with the biology underlying aortic dissection. GO terms with corrected *p* values < 0.05 and KEGG pathways with *p* values < 0.05 were defined as significant enrichments. The R packages ggplot2, enrichplot, and GO plot were used to create bubble and circle plots of GO and KEGG pathway annotations.

### 2.4. Construction of a Protein–Protein Interaction (PPI) Network

A protein–protein interaction (PPI) network based on the OSDEGs was constructed using STRING (version:12.0; https://string-db.org/), with the cut-off standard as a combined score of >0.9. Subsequently, the PPI networks were exported from STRING and imported into Cytoscape (https://cytoscape.org/). We ranked all the listed genes according to their degrees, which were calculated using PPI and co-expression networks. Molecular Complex Detection (MCODE) was used to identify significant modules in the PPI network.

### 2.5. Transcription Factor (TF) and miRNA Regulatory Network Analysis of OSDEGs

We submitted a list of OSDEGs to NetworkAnalyst (version: 3.0; https://www.networkanalyst.ca/). Potential transcription factors were predicted using the JASPAR database, and a network diagram was drawn. Relevant miRNAs were predicted using the TarBase database, and a network diagram was drawn [18,19,20].

### 2.6. Construction of Weighted Gene Co-Expression Network Analysis (WGCNA)

We conducted a Weighted Gene Co-expression Network Analysis (WGCNA) using the R package “WGCNA”, a method aimed at distinguishing modules of genes with high correlation, mapping interconnections between these modules and their corresponding relations with external traits of samples, and pinpointing possible biomarkers or therapeutic targets [21]. In this study, we applied WGCNA to identify the modules most relevant to AVF failure. Prior to this, we processed the sample data to exclude outliers. Following this, we used the “WGCNA” software package (version 1.72; https://cran.r-project.org/web/packages/WGCNA/index.html; accessed on 10 June 2022) to establish a correlation matrix. An optimal soft threshold is selected to convert the correlation matrix into an adjacency matrix that is subsequently used to create a topological overlap matrix (TOM). We leveraged the metric of phase dissimilarity based on the TOM to group genes with similar expression patterns into gene modules by employing average linkage hierarchical clustering. The module with the strongest correlation with the disease phenotype was selected as the key module for subsequent analysis, and the core genes in this module were extracted.

### 2.7. Machine Learning

The Least Absolute Shrinkage and Selection Operator (LASSO) logistic regression analysis is a method for data mining that, through the use of the L1-penalty (lambda), effectively sets the coefficients of non-significant variables to zero. This serves to highlight significant variables and build the most effective classification model [22]. Support Vector Machine-Recursive Feature Elimination (SVM-RFE) analysis is a governed machine learning method that identifies optimal core genes by removing the feature vectors generated by SVM [23]. Random Forest (RF) analysis is a machine learning method based on decision trees that evaluates the importance of variables by scoring each variable’s significance [24]. The diagnostic genes from OSDEG were separately evaluated using these three machine learning algorithms [25,26]. The intersection of the three algorithms was then determined to identify the intersecting genes.

### 2.8. Human Tissue Collection

Venous samples from both native and AVF patients were obtained from the Wuhan Union Hospital in Wuhan, China. Informed consent was obtained from all the patients. This study was approved by the Ethics Committee of the Tongji Medical College of Huazhong University of Science and Technology (Approval Number [2020] IEC-J (117)). This study was conducted in accordance with the International Ethical Guidelines for Biomedical Research Involving Human Subjects issued by the Council for the International Organization of Medical Sciences and complied with the guidelines set out in the Declaration of Helsinki. Subsequent immunohistochemical validation was performed on the paraffin sections of the collected native veins and AVF venous samples.

### 2.9. Animal Study

Male C57BL/6J mice (6–8 weeks old) were purchased from Vital River Laboratory Animal Technology Co., Ltd. (Beijing, China). The mice were housed under controlled conditions of 55 ± 5% humidity and 22 ± 2 °C under a 12 h light/dark cycle with ad libitum access to food and water. All animals underwent adaptive feeding for 3 days before further treatment.

An AVF was generated in each mouse after a partial nephrectomy to establish a chronic kidney disease (CKD) model. The mice were anesthetized with a mixture of ketamine (100 mg/kg body weight) and xylazine (10 mg/kg body weight) before surgery, and a CKD model was created by surgical ligation of the arterial blood supply to the upper pole of the left kidney, accompanied by removal of the right kidney, as previously described [27,28]. Mice that underwent surgery showed an increase in urea nitrogen and creatinine 4 weeks after nephrectomy. Twenty-eight days after the establishment of CKD, we connected the right common carotid artery to the right external jugular vein to create an AVF, as previously described [29,30]. The mice were divided equally into two groups. In one group, NS398 was injected intraperitoneally every other day at a dose of 3 mg/kg body weight, starting 3 days after the establishment of an AVF [31,32]. The remaining mice were injected with the same dose of vehicle as in the control group. Outflow venous samples were obtained near the anastomosis in different groups by euthanizing the mice on days 7 and 28 after surgery. The 7-day samples were used for frozen section detection of dihydroethidium (DHE) fluorescence and the preparation of tissue homogenates to measure oxidative stress via detection kits. Samples collected 28 days after AVF surgery were used to prepare paraffin sections for immunohistochemical staining and morphological analysis.

All experiments involving animal treatment were performed in accordance with the National Institutes of Health Guide for the Care and Use of Laboratory Animals, and the protocol was approved by the Ethics Committee of the Union Hospital affiliated with Huazhong University of Science and Technology (Approval Number 2636).

### 2.10. Doppler Ultrasound

Ultrasound observations were performed on the 28th day using a high-resolution Vevo 2100 microimaging system with an MX400 linear array transducer at 30 MHz (Visual Sonics, Toronto, ON, Canada). Peak velocity (PV) and mean velocity (MV) data were acquired using this ultrasound system. The wall shear stress (WSS) was calculated by the equation WSS = 4ηV/r, where η is blood viscosity, V is peak flow velocity (cm/s), and r is the radius (cm). The blood viscosity was assumed to be constant at 0.035 P. We calculated the AVF flow rate (mL/min) by the following equation: AVF flow rate = MV × π × D2 × 60/400 [33], where V is the mean MV and D is the diameter.

### 2.11. Serum Creatinine and Blood Urea

Serum creatinine and blood urea nitrogen levels were measured 28 days after CKD model creation by using a serum creatinine assay kit (AmyJet Scientific, Wuhan, China) and a urea assay kit (AmyJet Scientific, Wuhan, China).

### 2.12. Morphometric and Immunohistochemical Analysis

Tissue samples were initially preserved in 4% paraformaldehyde for 48 h and then transferred to 70% ethanol. They were processed through a sequence of varying ethanol solutions before being embedded in paraffin and then sectioned into 3-micrometer-thin slices. Following standard procedures, the slices were dewaxed, dehydrated, and repaired. The slices were blocked and incubated with the appropriate primary antibody and goat anti-rabbit/mouse immunoglobulin G reagents. Tissue sections underwent further treatment with horseradish peroxidase-labeled streptomyces ovalbumin protein and were then developed using diaminobenzidine. Hematoxylin was used as a counterstain. Gene expression was measured as the integrated optical density of areas stained yellow-brown using Image Pro Plus software (version 6.0, MEDIA Systems, Houston, TX, USA). The average optical density was obtained by dividing the integrated optical density by the area of target distribution. ImageJ software (version 1.5, National Institutes of Health, Bethesda, MD, USA) was used to analyze the intima and lumen areas of the outflow tract.

### 2.13. Statistical Analyses

Data were presented as mean ± SD. All data were analyzed using the GraphPad Prism 8 software (GraphPad Software, La Jolla, CA, USA). An analysis of variance with repeated measures, followed by a post-hoc Bonferroni correction or Student’s two-sided *t*-test, was used. For all comparisons, *p* values < 0.05 were considered statistically significant.

## 3. Results

### 3.1. Oxidative Stress Is Elevated in AVF Failure Patients and Identification of OSDEGs

Several studies have reported that oxidative stress plays an important role in the occurrence and development of AVF. First, we validated the upregulation of oxidative stress in the samples from patients with AVF failure.

Hematoxylin and eosin (HE) staining showed intimal thickening in the outflow tract of AVF failure (Figure 1A). The biomarkers 8-hydroxy-2 deoxyguanosine (8-OHdG) and 4-hydroxynonenal (4HNE) are important indicators of oxidative stress. Immunohistochemistry showed that the expression of 8-OHdG and 4HNE was significantly upregulated in the tissues of patients with AVF, in the hyperplastic intima and media, and this further suggested that oxidative stress was significantly increased in patients with AVF stenosis (Figure 1B–E).

The dataset is the only second-generation test dataset with a large sample size in the study of AVF that provides good support and helps in the study of AVF pathogenesis [16,34]. The results of the PCA analysis showed that the samples of the AVF and normal vein groups were well clustered and could be further analyzed (Figure 1F). A total of 694 differentially expressed genes were identified, and the numbers of upregulated and downregulated genes were determined, as shown in the volcano map (Figure 1G). By considering the intersection with the 1760 oxidative stress gene set obtained from the Genecard website, we identified 83 OSDEGs and drew a heat map of these differentially expressed genes (Figure 1H).

### 3.2. Functional Enrichment Analysis and Regulatory Network of OSDEGs

GO and KEGG enrichment analyses were performed based on the OSDEGs common to the native vein and AVF groups. In the KEGG pathway enrichment analysis, OSDEGs were particularly abundant in the PI3K−Akt signaling pathway, AGE−RAGE signaling pathway in diabetic complications, lipid, atherosclerosis, and other pathways (Figure 2A). GO enrichment analysis revealed that OSDEGs were significantly enriched in biological processes involved in responses to oxidative stress and steroid hormones. The DEGs also showed a certain degree of enrichment in cellular components, including the collagen-containing extracellular matrix and endoplasmic reticulum lumen. The DEGs were enriched in molecular functions such as receptor ligand activity and DNA-binding transcription activator activity (Figure 2B,C).

To explore the interplay among OSDEGs, PPI networks were built using the STRING tool with confidence >0.9 as the cut-off criterion (Figure 2D). Furthermore, according to the topological property analysis of the PPI network, FOS, JUN, and IL1B were the top three nodes with the greatest number of connections with other nodes (Figure 2E). Using MCODE, four clusters were identified in the network. These subnetworks may play an important role in AVF failure (Figure 2F–I).

Using the JASPAR database, seven transcription factors were finally obtained with degrees ≥2, and they were BRCA1, HINFP, FOXC1, E2F1, RELA, CREB1, and NR2F1 (Appendix A). Possible miRNAs were predicted by the TarBase database with 14 miRNAs of degree ≥2 (Appendix A).

### 3.3. WGCNA and Machine Learning-Identified PTGS2 as an Essential Biomarker

We explored the core set of genes affecting AD by using WGCNA. The first step was to cluster the samples and select a soft threshold (Figure 3A,B). The soft-threshold power was calibrated to 13 (scale-free R^2^ = 0.85). Subsequently, relevant core modules were screened. Among the obtained modules, we found that the blue and red modules had the highest correlation with the phenotype of AVF disease (Figure 3C,D). Therefore, we intercrossed the hub genes of these two modules with OSDEGs and obtained 70 intersection genes (Figure 3E).

Machine learning is often used to screen for disease-related biomarkers. In this study, 70 DEGs were further screened using three common machine learning approaches. First, seven genes were extracted from the DEIOSGs using the LASSO regression algorithm (Figure 3F). The SVM-RFE algorithm identified 13 genes (Figure 3G). Thirty genes were selected using the RF algorithm (Figure 3H). Finally, the three machine learning methods yielded one intersection gene, namely PTGS2 (Figure 3I).

### 3.4. Validation of PTGS2 and Immune Infiltration Analysis

First, we compared the expression changes of PTGS2 in different groups using GSE119296 and constructed an expression box plot. It was found that the expression of PTGS2 in the AVF group was greatly upregulated (Figure 4A). The GSE39488 dataset is an early AVF microarray dataset. Similar to another test dataset used in the present study, the GSE39488 dataset also showed the same trend (Figure 4B). To explore the diagnostic efficacy of PTGS2, we implemented a receiver operating characteristic (ROC) curve analysis in which hub genes with an area under the curve (AUC) value >0.7 were used as diagnostic markers. In the GSE119296 dataset, the AUC value was 0.979 for PTGS2 (Figure 4C). In the GSE39488 dataset, the AUC value was 0.958 for PTGS2 (Figure 4D). We further verified the expression level of PTGS2 in the tissues of these patients. PTGS2 fluorescence was significantly higher in the AVF group than that in the control vein group (Figure 4E,F).

Immune cell infiltration plays an important role in cardiovascular diseases. Therefore, the relationship between immune cell infiltration and PTGS2 expression was investigated. The bar chart shows the proportions of infiltrated immune cells (Figure 4G). Significant differences were observed between the four immune cell types (Figure 4H). Furthermore, we estimated the correlation efficiency between immune cells and PTGS2 to investigate their link and potential interaction. PTGS2 expression was significantly correlated with mast cell activation, neutrophils, T cells, CD4 memory resting cells, Tregs, CD4 native T cells, and resting mast cells (Figure 4I). The highest positive correlation coefficient was observed for mast cell activation, and the highest negative correlation coefficient was observed for resting mast cells (Figure 4J,K).

### 3.5. PTGS2 Inhibition Alleviates Lumen Stenosis in the AVF Mouse Model 

To test whether targeting PTGS2 could alleviate intimal hyperplasia and luminal stenosis in AVF, NS398, a common selective inhibitor of PTGS2, was used in animal experiments. To recreate the actual clinical situation and eliminate the influence of CKD on the experiment, we first constructed a CKD model. After the establishment of the CKD mouse model, urea nitrogen and creatinine levels were significantly upregulated (Appendix A).

The AVF model was subsequently established in these mice by creating end-to-side anastomosis of the cervical vessels (Figure 5A). This surgical method is similar to the current mainstream dialysis access construction methods. The use of NS398 significantly increased the outflow tract diameter 4 weeks after AVF, indicating improved lumen patency and higher blood flow (Figure 5B). HE staining showed that treatment with NS398 increased the size of the AVF outflow tract and decreased the neointimal area (Figure 5C–E). Masson’s trichrome staining showed that the use of NS398 also reduced the content of collagen fibers in the AVF specimens (Figure 5F,G).

### 3.6. Hemodynamic Changes after NS398 Administration Compared with Controls

Ultrasonography is often used to evaluate the patency of the AVF outflow tract in clinical practice. At 28 days after AVF establishment, the mean diameter of the outflow vein was significantly greater in the PTGS2 inhibitor group than in the control group (Figure 6A,B). In addition, the outflow veins showed a regular arterial flow signal with clearly visible peaks and troughs (Figure 6C). Blood flow velocity is an important indicator of AVF stenosis in clinical practice and animal studies. Blood flow velocity stenosis often indicates luminal stenosis and a poor prognosis [35,36]. Similarly, in our study, the peak flow rate was positively correlated with the inner diameter of the lumen (Figure 6D). In addition, the use of NS398 resulted in a substantial increase in peak flow velocity (Figure 6E). The mean and peak flow velocities exhibited similar trends (Figure 6F). In addition, the outflow vessel blood flow and WSS were calculated to assess AVF function. Low blood flow and WSS values are clinically and experimentally associated with AVF failure and intimal hyperplasia [37,38]. The use of NS398 increased the outflow (Figure 6G). However, in the present study, no significant difference in the WSS value was observed between the two groups (Figure 6H). To test whether targeting PTGS2 could alleviate intimal hyperplasia and luminal stenosis in AVF, NS398, a common selective inhibitor of PTGS2, was used in animal experiments. To recreate the actual clinical situation and eliminate the influence of CKD on the experiment, we first constructed a CKD model. After the establishment of the CKD mouse model, urea nitrogen and creatinine levels were significantly upregulated (Appendix A).

### 3.7. The Administration of NS398 Can Attenuate the Proliferation and Migration of Smooth Muscle Cells in AVF 

The main pathological change in vascular outflow tract hyperplasia is the massive proliferation of smooth muscle cells. Previous studies have shown that the use of NS398 at the cellular level affects the proliferation and migration of smooth muscle cells [13,14,15]. We further explored the specific effects of NS398 on AVF in animal tissues. We first confirmed that smooth muscle actin-positive cells were predominantly concentrated in the proliferatiIe intima fraction and that NS398 treatment substantially reduced alpha smooth muscle actin deposition (Figure 7A,B). Subsequently, we used immunohistochemical staining and found that NS398 significantly reduced the expression of collagen-1 and matrix metalloprotease-9 (MMP-9), indicating that cell migration ability and extracellular matrix secretion were decreased (Figure 7C–F). NS398 reduced proliferating cell nuclear antigen (PCNA) expression, suggesting that it attenuated cell proliferation (Figure 7G,H). Interestingly, there was no difference between the two groups, as determined by TUNEL staining (Figure 7I,J). These results indicate that PTGS2 inhibition occurs only by inhibiting the proliferation of smooth muscle cells and not by inducing apoptosis.

### 3.8. NS398 Can Reduce Oxidative Stress in AVF 

ROS levels were assessed by DHE staining. The administration of NS398 attenuated ROS production (Figure 8A,D). Similarly, 4NHE and 8-OHdG are common biomarkers of oxidative stress. The administration of NS398 attenuated the production of both of these substances, especially at the intimal site (Figure 8B,C,E,F). Subsequently, a kit was used to detect oxidative stress-related enzymes. NS398 reduced MDA content (Figure 8G) and increased CAT and SOD levels (Figure 8H,I).

## 4. Discussion

In this study, we verified the upregulation of oxidative stress in patient tissue sections. Subsequently, by analyzing the sequencing data from the AVF public database and the oxidative stress dataset, the related oxidative stress differential genes were identified, and the important signaling pathways and PPI networks enriched by these genes were analyzed. Simultaneously, with progress in various bioinformatics techniques, machine learning algorithms and WGCNA have become more established and have been widely used in the prediction of disease markers and therapeutic targets. In this study, we used the OSDEG gene collection combined with multiple machine learning (WGCNA) tools to identify key genes related to oxidative stress, namely PTGS2, and validated it using additional datasets. Analysis of the expression of key genes showed that PTGS2 expression was greatly increased in the disease group, and ROC curve analysis of the diagnostic value of key genes showed that this gene had excellent diagnostic efficacy.

We investigated the role of PTGS2 in the pathogenesis of intimal hyperplasia in AVF using both clinical benchmarks and animal models. PTGS2 expression was significantly higher in human specimens removed from stenotic hemodialysis AVFs than in controls. Histomorphometric analysis of outflow tract veins blocked by NS398 showed that the neointimal area was significantly reduced and the lumen size was increased compared to the control group. The proliferation of alpha smooth muscle antigen-positive smooth muscle cells is the basic pathological change of intimal hyperplasia in AVF. Smooth muscle cells can switch from a contractile to a synthetic morphology under multiple stimuli, including oxidative stress and inflammation. In this study, treatment with NS398 significantly reduced the expression of alpha smooth muscle actin-positive cells and increased the expression of PCNA-positive cells. Interestingly, no change in apoptosis was observed in this study, indicating that NS398 did not reduce the number of cells in the neointima through apoptosis. In addition, the expression of MMP-9 and collagen-1 was decreased in the PTGS2 inhibitor group, suggesting that the extracellular matrix secretion and cell migration abilities of smooth muscle cells were decreased.

Treatment with NS398 reduced oxidative stress in the outflow tract veins, as shown by ROS fluorescence staining and oxidative stress-related 4HNE and 8-OHdG immunohistochemical staining. This result was further validated using an associated oxidative stress product kit. These results suggest that PTGS2 inhibition could alter oxidative stress in the outflow tract. The initial discussion between AVF and oxidative stress came from the finding of increased oxidative stress and growth factor expression after AVF failure in human specimens [5]. Currently, a few antioxidative stress treatment strategies have been used in related AVF studies. Previous studies have mainly focused on reducing oxidative stress and enhancing the activity of endogenous antioxidant enzymes [39,40]. Unfortunately, these beneficial effects do not translate into positive results in clinical treatments. The findings of the present study can complement the failed antioxidative stress approaches for AVF. Compared to non-specific antioxidative stress clearance, targeting upstream-related genes may show better effects.

However, this study has several limitations. It only investigated the effect of targeting PTGS2 on intimal hyperplasia of AVF by using datasets, clinical samples, and AVF mouse model animal studies, and did not involve in vitro experiments. Many studies have investigated how PTGS2 affects smooth muscle cell proliferation and migration at the cellular level [11,12,41]. For example, Choi et al. found that inhibition of PTGS2 could inhibit the proliferation and migration of smooth muscle cells by increasing the expression of HO-1 [12]. The study by Lee et al. showed that silencing or pharmacological inhibition of PTGS2 significantly affected tumor necrosis factor alpha-induced phenotypic transformation of smooth muscle cells [41]. Zhang et al. found that PTGS2-derived prostaglandin E2 increases intimal hyperplasia after vascular injury by affecting the phenotypic switching of smooth muscle cells [11]. Therefore, the specific downstream molecules and signaling pathways of PTGS2 were not explored in detail in this study. Related experiments are currently underway. In addition, Western blotting experiments were not used to further verify the related molecules in this experiment. Since the neck outflow tract size in mice is approximately 0.5 mm in diameter, the total mass of a single outflow tract tissue sample is less than 5 mg, and this is extremely difficult during tissue protein extraction. Therefore, more relevant protein experiments should be conducted in rats or medium-sized animals.

In conclusion, our study identified PTGS2 as an important biomarker with a good diagnostic value for AVF failure. Targeted inhibition of PTGS2 plays an important role in improving intimal hyperplasia and blood flow patency in AVF animal models and affects the proliferation and migration of smooth muscle cells, as well as local oxidative stress in AVF.

## Figures and Tables

**Figure 1 antioxidants-13-00005-f001:**
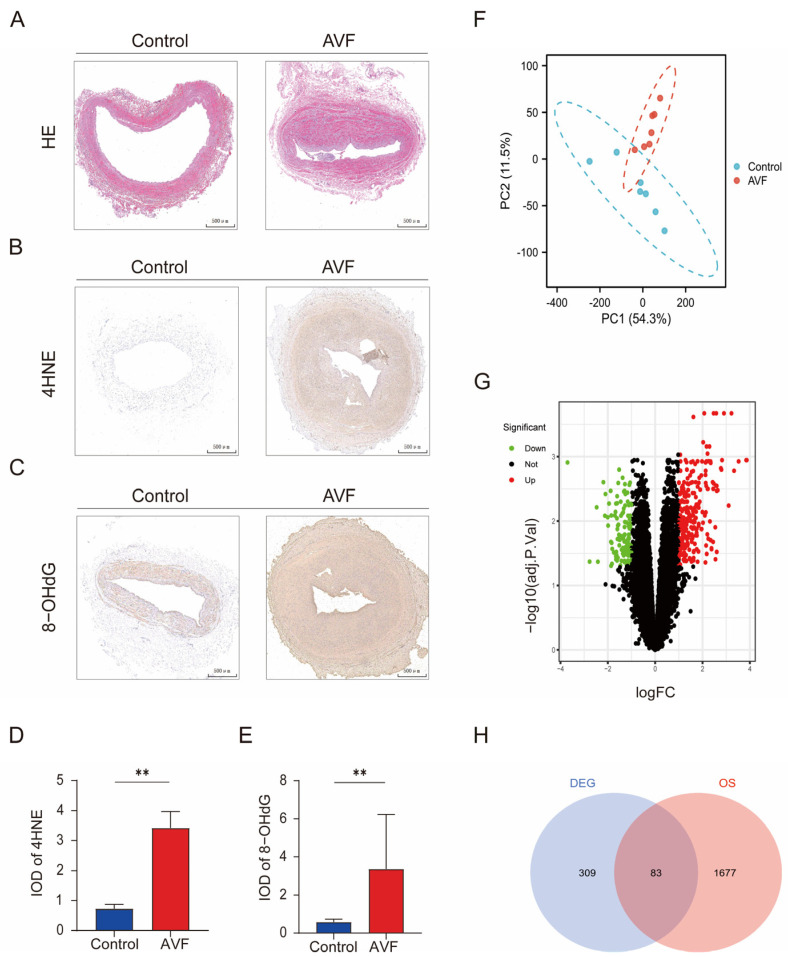
Oxidative stress elevated in AVF failure patients and identification of OSDEGs. (**A**) HE representative images of native vein (control) and failed (stenotic) AVF specimens. (**B**) Representative images of immunohistochemical staining for 4HNE in the native vein and AVF failure specimens. The scale bar is 500 µm. (**C**) Representative images of immunohistochemical staining for 8-OHdG in the native vein and AVF failure specimens. The scale bar is 500 µm. (**D**,**E**) The quantitative analysis of 4HNE and 8-OHdG in the native vein and AVF failure specimens (N = 3). Significant differences are indicated by ** *p* < 0.01. (**F**) PCA plot of GSE119296. (**G**) Volcano plot of DEGs in GSE119296. (**H**) Venn plots drawn for the DEGs and oxidative stress genes.

**Figure 2 antioxidants-13-00005-f002:**
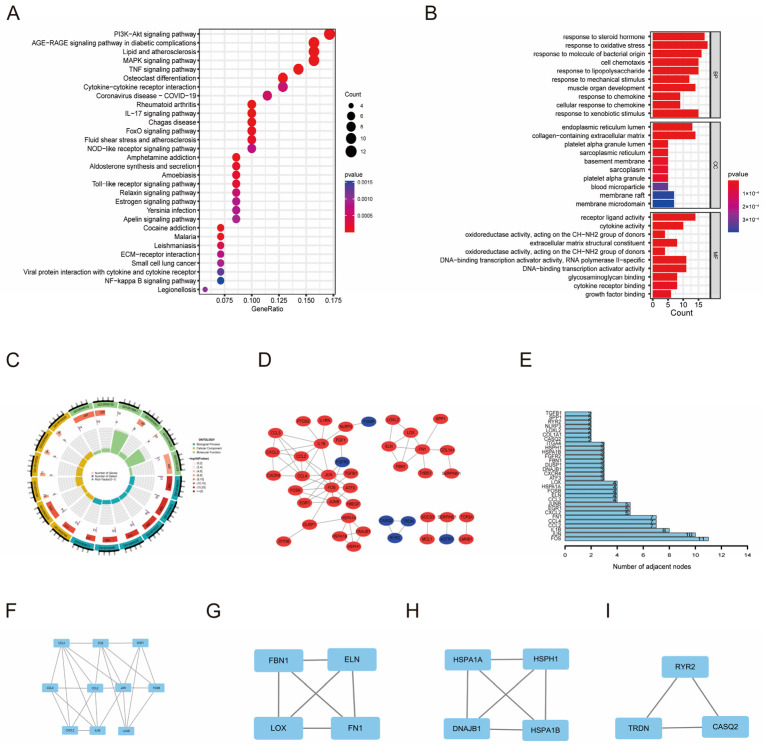
Enrichment analysis and PPI analysis of OSDEGs. (**A**) KEGG pathway enrichment analysis in OSDEGs. (**B**,**C**) GO pathway enrichment analysis in OSDEGs. (**D**) PPI network of the OSDEGs. (**E**) Hub genes generated by the PPI network. (**F**–**I**) Four subnetworks generated by MCODE.

**Figure 3 antioxidants-13-00005-f003:**
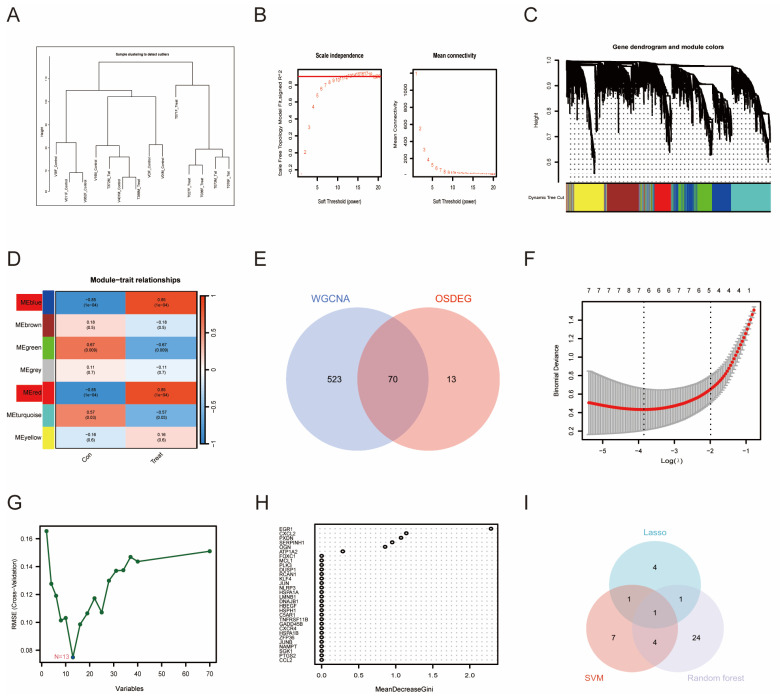
Screening diagnostic genes by overlapping gene analysis (WGCNA) and machine learning. (**A**) Samples from different groups were clustered; (**B**) choosing the best soft-threshold power; (**C**) cluster plot of different genes by the WGCNA; (**D**) 7 modules revealed by the WGCNA, where blue and red modules have the highest correlation with the phenotype of the disease; (**E**) Venn plots drawn for the OSDEGs and WGCNA hub genes; (**F**) LASSO regression algorithm; (**G**) SVM-RFE algorithm; (**H**) RF algorithm; and (**I**) Venn diagrams for three algorithms.

**Figure 4 antioxidants-13-00005-f004:**
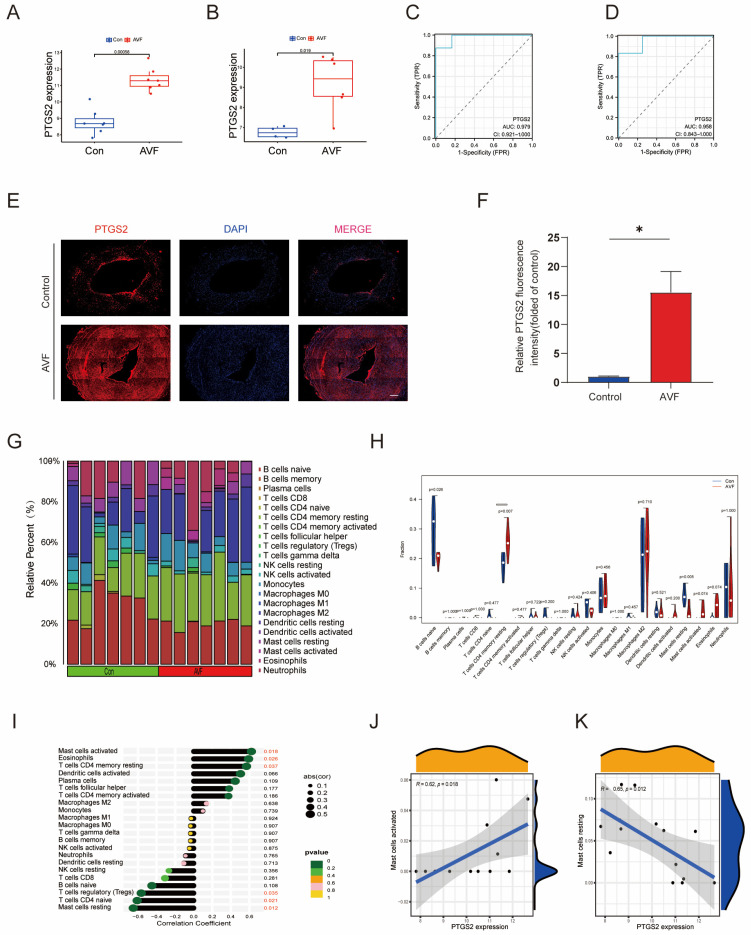
Validation of PTGS2 and immune infiltration analysis. (**A**) Expression of PTGS2 in the GSE119296 dataset; (**B**) expression of PTGS2 in the GSE39488 dataset; (**C**) PTGS2 in the GSE119296 dataset were analyzed using ROC curve; (**D**) PTGS2 in the GSE39488 dataset were analyzed using ROC curve; (**E**) representative images of immunofluorescence staining of PTGS2 in the native vein and AVF failure specimens; the scale bars are 200 μm; (**F**) quantitative analysis of PTGS2 in the native vein and AVF failure specimens (N = 3); (**G**) bar charts of immune cell infiltration in GSE119296; (**H**) box plot of immune cell infiltration differences in GSE119296; (**I**) lollipop plot of the association of immune cells with AVF failure; (**J**) highest positive correlation coefficient with PTGS2 observed for mast cell activation; (**K**) highest negative correlation coefficient with PTGS2 observed for mast cell resting. Each bar represents the mean ± SD. Significant differences are indicated by * *p* < 0.05.

**Figure 5 antioxidants-13-00005-f005:**
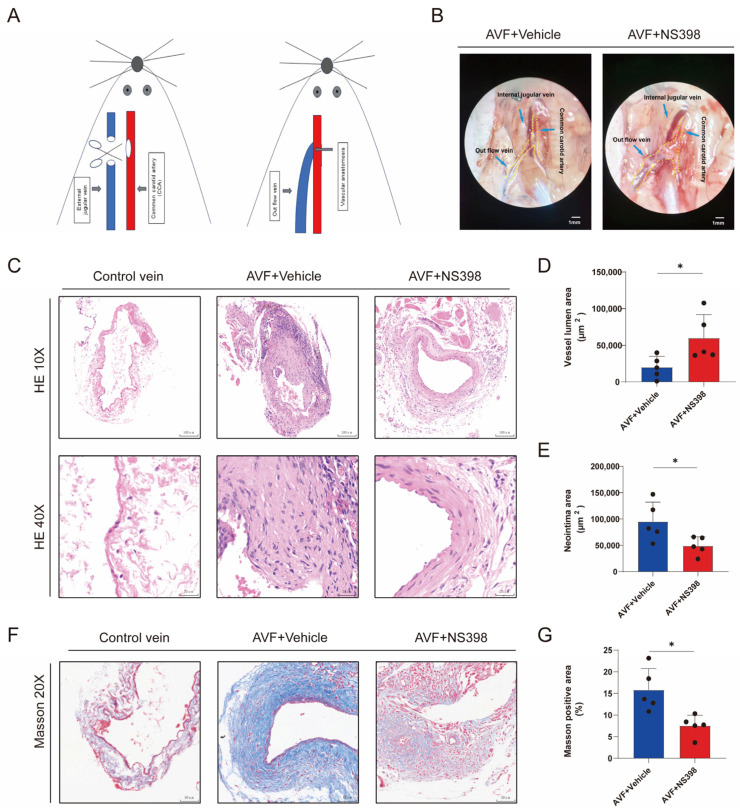
PTGS2 inhibition alleviates lumen stenosis in the AVF mouse model. (**A**) Schematic representation of the AVF model construction, where the arrows in the figure indicate the common carotid artery and external jugular vein used for anastomosis, respectively; (**B**) representative picture of outflow tract veins 28 days after AVF construction in different groups, where the yellow dotted line shows the common carotid artery and the outflow tract of AVF; the scale bar is 1 mm; (**C**) representative hematoxylin and eosin-stained sections for control vein, AVF, and NS398-treated AVF at day 28, respectively; the scale bars are 100 and 25 μm, respectively; (**D**) administration of NS398 significantly increased the lumen area (N = 5 for each group); (**E**) significant reduction in the average neointimal area in NS398 treated group compared with vehicle group (N = 5 for each group); (**F**) representative Masson stained sections for control vein, AVF, and NS398-treated AVF at day 28, respectively; the scale bars are 100 and 25 μm, respectively; (**G**) administration of NS398 significantly decreased the Masson positive area (N = 5 for each group). Each bar represents the mean ± SD. Significant differences are indicated by * *p* < 0.05; ns, not significant. HE, hematoxylin and eosin.

**Figure 6 antioxidants-13-00005-f006:**
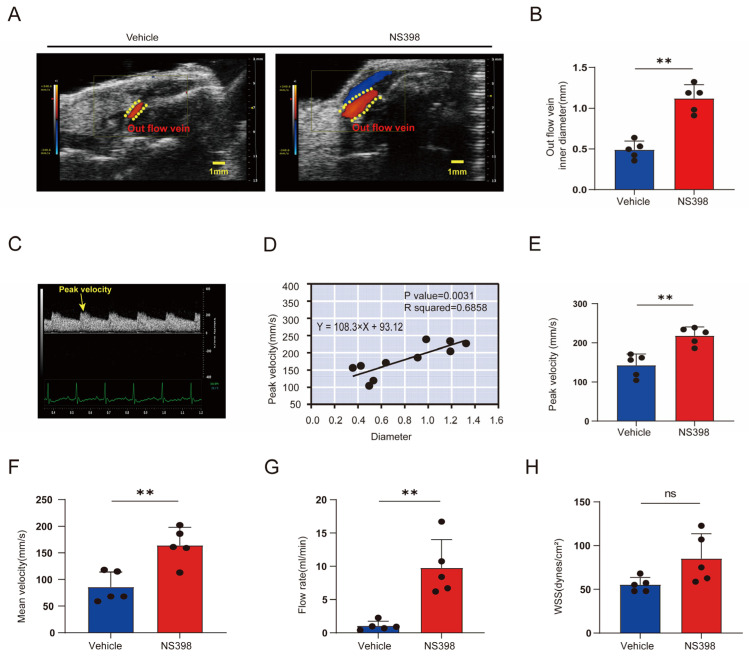
Hemodynamic changes after NS398 administration compared with controls. (**A**) Representative Doppler image of the AVF outflow tract; (**B**) application of NS398 can greatly increase the diameter of the outflow vein; (**C**) representative image of blood flow velocity; (**D**) linear regression of peak flow velocity versus vessel inner diameter; (**E**) application of NS398 increased the peak flow velocity in the outflow tract; (**F**) application of NS398 increased the mean outflow velocity; (**G**) application of NS398 increased outflow tract blood flow; (**H**) statistical plot of wall shear stress between different groups. Significant differences are indicated by ** *p* < 0.01; ns, not significant.

**Figure 7 antioxidants-13-00005-f007:**
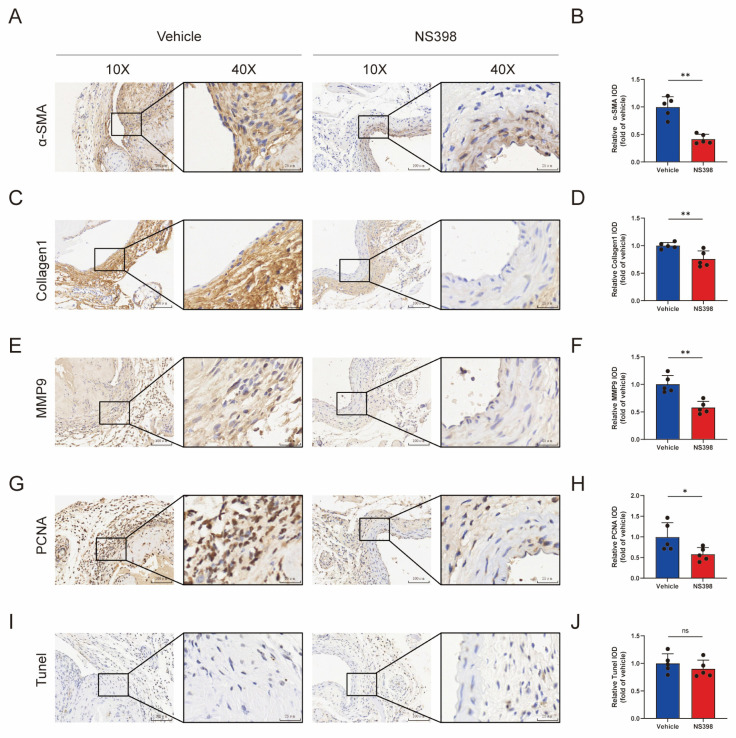
The administration of NS398 can attenuate the proliferation and migration of smooth muscle cells in AVF. (**A**,**B**) Representative images of immunohistochemical staining for a-SMA as well as statistical plot; (**C**,**D**) representative images of immunohistochemical staining for collagen-1 as well as statistical plot; (**E**,**F**) representative images of immunohistochemical staining for MMP9 as well as statistical plot; (**G**,**H**) representative images of immunohistochemical staining for PCNA as well as statistical plot; (**I**,**J**) representative images of immunohistochemical staining for Tunel as well as statistical plot. The scale bars are 100 and 25 μm, respectively. Each bar represents the mean ± SD. Significant differences are indicated by * *p* < 0.05 and ** *p* < 0.01. ns, not significant. N = 5 for each group.

**Figure 8 antioxidants-13-00005-f008:**
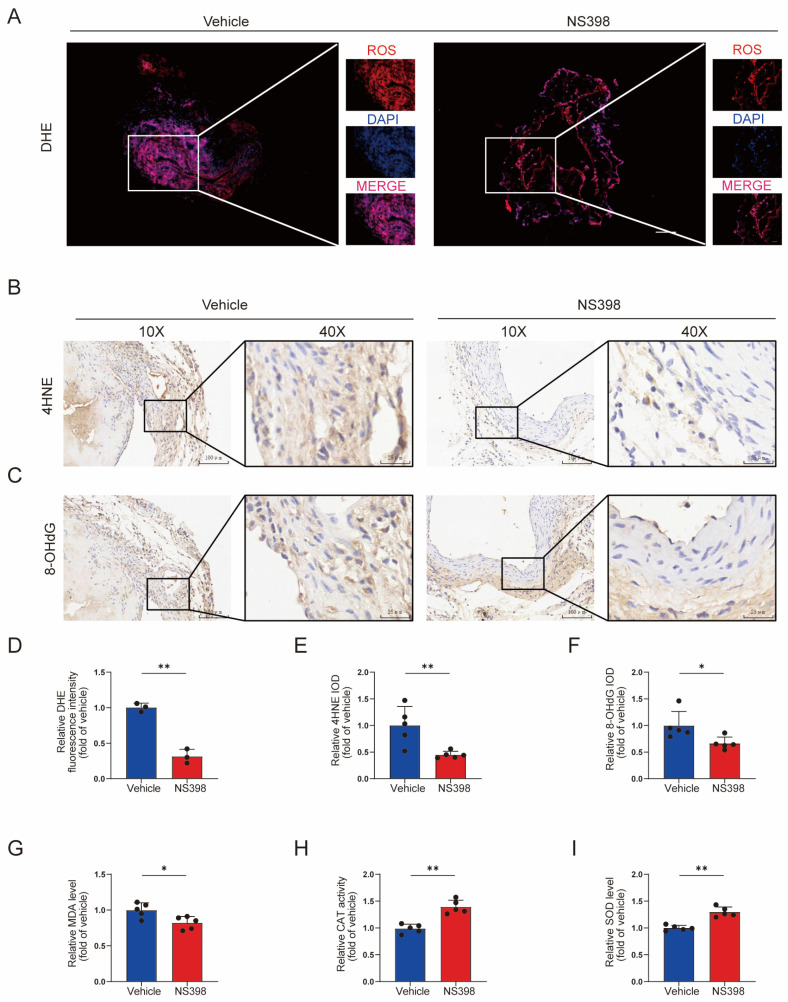
NS398 can reduce oxidative stress in AVF. (**A**) Representative fluorescence image for DHE of the vehicle group and the NS398 group. The scale bars are 100 and 25 μm, respectively. (**B**,**C**) Representative immunohistochemical images of 4HNE and 8OHDG. The scale bars are 100 and 25 μm, respectively. (**D**–**F**) Quantitative statistical analysis of DHE, 4HNE, and 8-OHdG. (**G**–**I**) Quantitative statistics of MDA, CAT, and SOD kits. Each bar represents the mean ± SD. Significant differences are indicated by * *p* < 0.05 and ** *p* < 0.01. N = 5 for each group.

## Data Availability

The datasets used and/or analyzed during the current study are available from the corresponding author on reasonable request.

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
