# Peer review of "Identification and Validation of PTGS2 Gene as an Oxidative Stress-Related Biomarker for Arteriovenous Fistula Failure"

_antioxidants, 2023, doi:10.3390/antiox13010005_

Round 1

Reviewer 1 Report

Comments and Suggestions for Authors

Dear authors,

Let me start by congratulating you on the article. I found the study to be interesting, but I think there are a few issues that should be corrected before this manuscript is published.

Abstract

The abstract should contain numerical data.

Introduction

Some corrections are required. Line 34 – “Arteriovenous fistulas (AVFs) were developed more than half a century ago………”. Please rephrase, It is not clear enough.

Materials and Methods

Lines 147 – 155: In the "Human Tissue Collection" chapter, data related to in vivo experiments on mice are included. Please correct and include this data in a separate chapter.

Results

The images are generally of inadequate quality. The majority of the figures include very small pictures and graphics that make them difficult to comprehend.

Although histopathological images are interesting they also have a visual aid issue because they are far too small. All histological staining images are sub-standard. It is suggested to enhance those and include high resolution images.

Author Response

Dear Reviewer,

We are very grateful for taking a careful look at our article and gave us helpful comments that are crucial to our research. We have carefully addressed all the concerns and made corresponding changes to the manuscript. We hope our manuscript can meet your publishing requirements on Antioxidants.

Sincerely yours,

Weici Wang (on behalf of all authors)

Comment #1: Abstract

The abstract should contain numerical data.

Reply 1: Thank you very much for your insightful advice. We added the main numerical data results “Based on 83 oxidative stress-related differentially expressed genes, we identified the important pathways related to oxidative stress. PTGS2 may have diagnostic and therapeutic efficacy for AVF failure. We further confirmed this finding using clinical specimens and validation datasets. The animal experiments illustrated that NS398 administration could reduce neointimal area (average decrease: 49%) and improve peak velocity (average increase: 53%).” in the Abstract to make the results more concrete (see Page 1, line 24-28).

Comment #2: Introduction

Some corrections are required. Line 34 – “Arteriovenous fistulas (AVFs) were developed more than half a century ago………”. Please rephrase, It is not clear enough.

Reply 2: Thank you very much for your suggestion. We have modified our text as advised to make it clearer. “In 1966, Brescia et al. succeeded in inventing a mature arteriovenous fistula (AVF) by surgically connecting an artery to a vein, thereby providing end-stage renal disease (ESRD) patients with dialysis access capable of delivering optimal flow.” (see Page 1, line 35-37).

Comment #3: Materials and Methods

Lines 147-155: In the "Human Tissue Collection" chapter, data related to in vivo experiments on mice are included. Please correct and include this data in a separate chapter.

Reply 3: Thank you so much for your careful check. We have corrected this by removing the data related to in vivo experiments on mice from the chapter "Human Tissue Collection" and placing it in the next separate chapter "Animal study" (see Page 4, line 146-150 and 170-173).

Comment #4: Results

The images are generally of inadequate quality. The majority of the figures include very small pictures and graphics that make them difficult to comprehend.

Although histopathological images are interesting they also have a visual aid issue because they are far too small. All histological staining images are sub-standard. It is suggested to enhance those and include high resolution images.

Reply 4: Thank you very much for your suggestion. We've increased the image resolution and reinserted them in the text. We will also upload the original figures to the system.

Reviewer 2 Report

Comments and Suggestions for Authors

The authors prepared the article in an exemplary manner. I mean the topic being discussed and how it was planned to be assessed. The analyses carried out are very thoughtful, extremely accurate, and cover all aspects, which is extremely important in research in the bio-medical area. However, I have a few reservations about the graphics. I believe that the engravings, especially Figs. 2 - fig. 4 - are entirely illegible in places and should be corrected. Maybe only some of the most valuable could stay, and the rest could be uploaded in the supplementary materials. An exciting addition would also be a diagram presenting the main results of the study - it could increase readers' interest, as well as facilitate understanding of the topic of oxidative stress, which is a challenge for many scientists. That's why I recommend a minor revision.

Author Response

Dear Reviewer,

We are very grateful for taking a careful look at our article and gave us helpful comments that are crucial to our research. We have carefully addressed all the concerns and made corresponding changes to the manuscript. We hope our manuscript can meet your publishing requirements on Antioxidants.

Sincerely yours,

Weici Wang (on behalf of all authors)

Comments: The authors prepared the article in an exemplary manner. I mean the topic being discussed and how it was planned to be assessed. The analyses carried out are very thoughtful, extremely accurate, and cover all aspects, which is extremely important in research in the bio-medical area. However, I have a few reservations about the graphics. I believe that the engravings, especially Figs. 2 - fig. 4 - are entirely illegible in places and should be corrected. Maybe only some of the most valuable could stay, and the rest could be uploaded in the supplementary materials. An exciting addition would also be a diagram presenting the main results of the study - it could increase readers' interest, as well as facilitate understanding of the topic of oxidative stress, which is a challenge for many scientists. That's why I recommend a minor revision.

Reply: We greatly appreciate and agree with the reviewer’s assessment. As the reviewer suggested, we have adjusted the resolution of all figures and we added "Figure S1" in the supplementary materials to better present the main content of the study. (see Page 2, line 75-76).

Reviewer 3 Report

Comments and Suggestions for Authors The paper is very good and does not need special criticism. For this reason I was very brief.   The physiopathology of  arteriovenous fistula failure is not well known. The paper by Ke Hu et al brings new information based on lot of complementary methods It shows clearly that PTGS2 gene is an oxidative 2 stress-related biomarker for arteriovenous fistula failure This finding is very important since this gene could be considered as a biomarker of this pathology. In addition, this gene could be used for the identification of anti-oxidant therapies which remains to be defined. May be PTSG2 could also be used in gene therapy in the future.

I consider not only that the paper is clear well written and well illustrated but I also consider it has a great medical potential.

Author Response

Dear Reviewer,

We are very grateful for taking a careful look at our article and gave us helpful comments that are crucial to our research. We have carefully addressed all the concerns and made corresponding changes to the manuscript. We hope our manuscript can meet your publishing requirements on Antioxidants.

Sincerely yours,

Weici Wang (on behalf of all authors)

Comments: The paper is very good and does not need special criticism. For this reason I was very brief. The physiopathology of arteriovenous fistula failure is not well known. The paper by Ke Hu et al brings new information based on lot of complementary methods It shows clearly that PTGS2 gene is an oxidative 2 stress-related biomarker for arteriovenous fistula failure This finding is very important since this gene could be considered as a biomarker of this pathology. In addition, this gene could be used for the identification of anti-oxidant therapies which remains to be defined. May be PTSG2 could also be used in gene therapy in the future.

I consider not only that the paper is clear well written and well illustrated but I also consider it has a great medical potential.

Reply: We greatly appreciate and agree with the reviewer’s assessment and it's a great encouragement to us.

Round 2

Reviewer 1 Report

Comments and Suggestions for Authors

I believe that in this form the manuscript can be accepted for publication.

Reviewer 2 Report

Comments and Suggestions for Authors

I think the article is excellent. The authors responded to all comments correctly, and in its current version, the article fully deserves acceptance and publication in this renowned journal.